# A Resistance-Based Microfluidic Chip for Deterministic Single Cell Trapping Followed by Immunofluorescence Staining

**DOI:** 10.3390/mi13081272

**Published:** 2022-08-07

**Authors:** Xiange Sun, Bowen Li, Wenman Li, Xiaodong Ren, Ning Su, Ruoxu Li, Jinmi Li, Qing Huang

**Affiliations:** Department of Laboratory Medicine, Daping Hospital, Army Medical University, Chongqing 400042, China

**Keywords:** microfluidic chip, single-cell trapping, immunofluorescence staining, least flow resistance

## Abstract

Microchips are fundamental tools for single-cell analysis. Although various microfluidic methods have been developed for single-cell trapping and analysis, most microchips cannot trap single cells deterministically for further analysis. In this paper, we describe a novel resistance-based microfluidic chip to implement deterministic single-cell trapping followed by immunofluorescence staining based on the least flow resistance principle. The design of a large circular structure before the constriction and the serpentine structure of the main channel made the flow resistance of the main channel higher than that of the trapping channel. Since cells preferred to follow paths with lower flow resistance, this design directed cells into the capture sites and improved single-cell trapping efficiency. We optimized the geometric parameters using numerical simulations. Experiments using A549 and K562 cell lines demonstrated the capability of our chip with (82.7 ± 2.4)% and (84 ± 3.3)% single-cell trapping efficiency, respectively. In addition, cells were immobilized at capture sites by applying the pulling forces at the outlet, which reduced the cell movement and loss and facilitated tracking of the cell in real time during the multistep immunofluorescence staining procedure. Due to the simple operation, high-efficiency single-cell trapping and lower cell loss, the proposed chip is expected to be a potential analytical platform for single tumor cell heterogeneity studies and clinical diagnosis.

## 1. Introduction

Since tumor cells always exhibit heterogeneity in both phenotypes and genotypes [1,2,3], cell population studies may mask the behavior of individual cells, which results in the loss of specific information. The cell heterogeneity has attracted considerable attention and a lot of researchers are working on the cell proteomics, genomics and metabolomics research at the single-cell level [4,5,6]. Single-cell analysis helps people to deeply understand intercellular differences, and obtain more comprehensive and accurate biological information, which facilitates the development of cancer research, drug screening and disease treatment.

To successfully isolate a single-cell from cell populations for further biochemical assays, various passive and active microfluidic methods have been developed for single-cell trapping and analysis benefiting from miniaturization and precise fluid manipulation [7,8,9,10,11,12]. Active methods showed great advantages in precise single-cell trapping due to the utilization of external force fields such as valve-driven [13,14], optical [15,16], acoustic [17,18], magnetic [19,20], electrical [21,22,23], and so on. However, active methods suffered from several problems including the complicated and multilayer structure fabrication, expensive and bulky external instruments and low throughput. Different from the active methods, passive microfluidic approaches achieved single-cell trapping based on the clever design of chip structures, such as micropatterns [24], microwells [25,26,27] and droplets [28,29]. However, single-cell trapping using the microwells and droplets followed the Poisson distribution, which resulted in random and inefficient single-cell trapping. Therefore, it was necessary to develop novel microfluidic chips with high trapping efficiency to meet the need of single-cell trapping and analysis.

It was reported that the deterministic single-cell trapping device could achieve high-efficiency single-cell trapping and analysis [30,31,32] because the cells could be captured one by one and be tracked in real time for multistep biochemical analysis, which was suitable for trapping rare cells such as circulating tumor cells. The least flow resistance microfluidic chip could achieve deterministic single-cell trapping relying on the dynamic variation of flow resistance inside the microchannels. Cells flowed along the path with minimum resistance and were captured sequentially. Owing to the simplicity of structure and operation, resistance-based microfluidic methods have been utilized in several studies [33,34,35,36]. For instance, Jin et al. [36] reported a passive least flow resistance microfluidic device, of which the core trapping structure consisted of a series of concatenated T and inverse T junction pairs, and 90% single-cell trapping efficiency was obtained. However, in order to realize single-cell trapping based on the least flow resistance principle, a longer main channel was usually needed to produce higher flow resistance than the trapping channel, which inevitably increased the chip area and cell loading time. Therefore, there was still a challenge to develop a chip with higher spatial and temporal efficiency.

In this paper, we report a resistance-based microfluidic chip with an array of trapping units, in which single-cell trapping and immunofluorescence staining were conducted in a deterministic manner. In order to make the flow resistance of the main channel higher than that of the trapping channel, we designed a large circular structure before the constriction to decrease the flow resistance of the trapping channel, and designed the serpentine structure to increase flow resistance of the main channel. Based on the least flow resistance principle, cells flowed into the capture sites and high-efficiency single-cell trapping was achieved. After single-cell trapping, immunofluorescence staining was carried out by applying the pulling forces at the outlet to deliver cell suspensions and staining reagents into the chip sequentially, which immobilized cells at capture sites and reduced cell movement and loss during the multistep immunofluorescence staining procedure. The chip enabled both single-cell trapping and immunofluorescence staining analysis in passive hydrodynamic deterministic orders by proper design of the microchannel structures and provided a novel strategy for single-cell trapping and analysis.

## 2. Materials and Methods

### 2.1. Design of the Single-Cell Trapping Chip

The proposed single-cell trapping chip consisted of 4 × 100 single-cell trapping units, and all the microchannels shared a joint inlet and outlet (Figure 1A). The detailed channel structures are shown in Figure 1B,C. The height H of the microfluidic channels was 25 μm. From point A to B, cells had two flow paths. The trapping channel consisted of a large circular structure and a constriction. The large circular structure was designed before the constriction to decrease the flow resistance of the trapping channel. The main channel was composed of the straight channel and serpentine structure. Compared with previous reports [33,34,35,36], in which the main channel was composed of the straight channel alone, the design of the serpentine structure improved the flow resistance of the main channel, which made the flow resistance of the main channel higher than that of the trapping channel, without a longer main channel or larger chip area.

### 2.2. Fabrication of the Single-Cell Trapping Microfluidic Chip

The microfluidic chip was designed using AutoCAD software and fabricated based on soft photolithography. The master mold was processed by Suzhou Hanguang Micro Nano Technology Co., Ltd. Sylgard 184 Elastomer Kit (Dow Corning, Midland, MI, USA) was used for the fabrication of the PDMS microfluidic chip. After 10:1 elastomer base and curing agent were mixed and degassed, the PDMS solution was poured into the master mold and cured at 80 °C for 30 min in an oven. After the cured PDMS layer was peeled from the mold, the inlet and outlet holes were punched manually using a puncher with a 1.4 mm external diameter. Finally, the PDMS was bonded to a glass substrate irreversibly via oxygen plasma treatment.

### 2.3. Numerical Simulations

In order to achieve highly efficient hydrodynamic single-cell trapping, COMSOL Multiphysics 5.2 a was utilized to optimize key geometric parameters of the microfluidic channels. The simulations assumed an uncompressed Newtonian fluid with the properties of water at room temperature and a no-slip boundary condition was applied on the channel surface. The Laminar Flow module and Normal mesh were used. The inlet velocity and outlet pressure were set as 1 μL/min and zero, respectively. The flow velocity was obtained by solving the Navier-Stokes equation and continuity equation. When simulating the flow velocity after cell trapping, cells were assumed as solid spheres in contact with the constriction. After simulating the flow velocity distribution, the volumetric flow rate of the trapping (i.e., *Q*_1_) and main channels (i.e., *Q*_2_) were obtained by the integral of the velocity over the cross section. Afterwards, we calculated the *Q* ratio values between trapping and main channels (i.e., *Q*_1_*/Q*_2_). Finally, we studied the effects of geometric parameters on the *Q* ratio values.

### 2.4. Cell Culture and Preparation

Adhesive lung cell line A549 was cultured in F12 K medium (Invitrogen, Waltham, MA, USA) supplemented with 10% (*v*/*v*) fetal bovine serum (FBS, Gibco, Waltham, MA, USA), 100 μg/mL streptomycin and 100 U/mL penicillin (Invitrogen, Waltham, MA, USA). Suspension cell line K562 was cultured using the RMPI 1640 medium (Hyclone, Logan, UT, USA) supplemented with 10% FBS and 100 μg/mL streptomycin and 100 U/mL penicillin. All cells were grown at 37 °C in 5% CO_2_ humidified atmosphere. Adherent A549 cells were detached from the cell culture flask (Corning, NY, USA) with 1.0 mL of 0.05% trypsin containing EDTA (Hyclone, Logan, UT, USA). Then, the F12 K medium with 10% FBS was added to deactivate trypsin. Finally, cells were washed with phosphate buffered saline (PBS, Hyclone, Logan, UT, USA) three times by centrifuging at 800 rpm for 3 min and re-suspended in PBS. The cell concentrations were analyzed using the Cell Counter (Thermal, CA, USA) and diluted to certain concentrations.

### 2.5. Single-Cell Trapping and Immunofluorescence Staining Analysis

To reduce cell movements and loss during the multistep immunofluorescence staining procedure, single-cell trapping and immunofluorescence staining were carried out by applying a pulling force at the outlet with a syringe pump to deliver the fluid into the chip, as shown in Figure 2. The outlet of the chip was connected to a syringe through the Teflon capillary tube. Pipette tips with a certain volume of solution were inserted into the inlet to deliver cell suspension and staining reagents into the chip. Prior to carrying out the single-cell trapping experiment, the chip was incubated with 3% bovine serum albumin (BSA, Sigma-Aldrich, Shanghai) to block channel surfaces and reduce cell adhesions. Then, 6 μL of cell suspension with 50 cells/μL of concentration was delivered into the chip. The withdraw mode of the syringe pump (Harvard, MA, USA) was used at a flow rate of 0.2 μL/min. After cell trapping, PBS was used to wash the channel.

Subsequently, immunofluorescence staining was conducted, in which various staining reagents were delivered into the chip sequentially, as shown in Figure 2A. Briefly, the trapped cells were fixed with 4% paraformaldehyde (PFA) at 0.1 μL/min for 15 min firstly, and then permeabilized with 3% BSA containing 0.1% TritonX-100 (Sigma-Aldrich, Shanghai) at 0.1 μL/min for 10 min. In the next step, cells were immunostained with 1:25 cytokeratin monoclonal antibodies conjugated with fluorescein isothiocyanate (CK-FITC, Miltenyi, Germany) at 0.1 μL/min for 90 min. Finally, cells were stained with 10 μg/mL 4′, 6-diamidino-2-phenylindole (DAPI, Beyotime, Shanghai, China) at 0.1 μL/min for 10 min. After staining, channels were washed with PBS and the images of single-cell trapping and immunofluorescence staining were observed with an inverted fluorescence microscope (Olympus, Tokyo, Japan).

### 2.6. Measurement of Cell Viability

Live and dead cells could be rapidly distinguished with the help of trypan blue staining based on the integrity of the cell membrane. The cell membrane of dead cells was incomplete and the permeability is increased so that trypan blue could penetrate into the cells and stain cells blue, while live cells were not stained due to the complete cell membrane. After cell trapping, 0.4% trypan blue solution (Invitrogen, Waltham, MA, USA) was delivered into the chip at 0.1 μL/min for 10 min. Then we counted the number of live cells and dead cells, respectively, under the inverted microscope. The cell viability was calculated by comparing the number of live cells to the total number of live cells and dead cells.

## 3. Results and Discussion

### 3.1. Principle of Single-Cell Trapping

In this study, a novel strategy was proposed to carry out deterministic single-cell trapping followed by immunofluorescence staining based on the least flow resistance principle. We designed a large circular structure just before the constriction to effectively decrease the flow resistance of the trapping channel, and designed a serpentine structure on the main channel to increase both the length and flow resistance of the main channel, which enabled the main channel to produce a higher flow resistance than the trapping channel. Figure 1D showed the cells’ movement paths in the channels. Before cell trapping, since cells preferred to flow along the paths having lower resistance, cells would flow along the trapping channel, and then were captured at the capture sites. After cell trapping, the flow resistance of the trapping channel increased significantly, which caused the subsequent cell to flow along the main channel and be captured by the next empty capture site. The proposed single-cell trapping chip did not need a longer main channel to produce higher flow resistance than the trapping channel, which reduced the chip area and sample loading time.

### 3.2. Theoretical Modeling of Single-Cell Trapping

The least flow resistance principle for single-cell trapping required that the flow resistance *R_2_* of the main channel should be higher than the flow resistance *R_1_* of the trapping channel. According to the Hagen-Poiseuille law [37] Δ*p* = *R*_hyd_*Q*, where Δ*p* was the difference of pressure from point A to B, and *Q* was the volumetric flow rate, the flow rate *Q_1_* of the trapping channel should be higher than the volumetric flow rate *Q_2_* of the main channel, namely *Q_1_*/*Q_2_* > 1. According to the Darcy-Weisbach law [38] to a rectangular channel, we obtained the expression:(1)$$△p=\frac{C(\alpha )}{32}\frac{\mu LQP^{2}}{A^{3}}$$

Here, *A* and *P* were the cross-sectional area and perimeter of the channels, and *Q*, *L*, *μ* were the volumetric flow rate, path length and fluid viscosity, respectively. The aspect ratio α was either height/width or width/height such that 0 ≤ α ≤ 1. *C (**α)* was the product of the Darcy friction factor *f* and Reynolds number *R_e_*,
(2)$$C\left(\alpha \right)=f\times Re=96\times \left(1-1.3553\cdot \alpha +1.9467\cdot \alpha^{2}-1.7012\cdot \alpha^{3}+0.9564\cdot \alpha^{4}-0.2537\cdot \alpha^{5}\right)$$

Because the pressure drops were the same between the main and trapping channels from point *A* to *B*, we ignored minor losses at bends, and widening and narrowing regions, and obtained the following expressions:(3)$$\frac{Q_{1}}{Q_{2}}=\frac{C(\alpha_{2})}{C(\alpha_{1})}\cdot {\left(\frac{A_{1}}{A_{2}}\right)}^{3}\cdot {\left(\frac{P_{2}}{P_{1}}\right)}^{2}\cdot \left(\frac{L_{2}}{L_{1}}\right)=\frac{C(\alpha_{2})}{C(\alpha_{1})}\cdot {\left(\frac{W_{1}}{W_{2}}\right)}^{3}\cdot \frac{{\left(W_{2}+H\right)}^{2}}{{\left(W_{1}+H\right)}^{2}{\left(2R+H\right)}^{2}}\cdot (\frac{L_{2}}{L_{1}})>1$$

Obviously, this final expression was only related to the width *W_1_* and length *L_1_* of the constriction, the radius *R* of the circular structure, the width *W*_2_ of main channel and the height *H* of channel, implying that the proper chip design played an important role in the realization of single-cell trapping.

### 3.3. Numerical Simulations of Single-Cell Trapping

The finite element simulations were used to study the influences of geometric parameters on the *Q* ratio values. Firstly, we studied the relationships between *L*_1_, *W*_1_, *R*, *W*_2_ and *Q* ratio values, respectively, in which when a parameter was studied, the other three parameters kept constant values and the flow rate was fixed at 0.1 μL/min. As shown in Figure 3, the red line represented the *Q* ratio values between the trapping and main channel before cell trapping, and the black line represented the *Q* ratio values after cell trapping. It was found that the increases of *W*_1_ and *R* led to the increases of *Q* ratio while the decreases of *L*_1_ and *W*_2_ made the *Q* ratio increase. Moreover, it worth noting that *W*_1_ and *W*_2_ had obvious effects on *Q* ratio compared with *L*_1_ and *W*_2_, implying that the design of geometric parameters *W*_1_ and *W*_2_ had important effects on the single-cell trapping. Considering that *Q*_1_/*Q*_2_ > 1 before cell trapping and *Q*_1_*/Q*_2_ < 1 after cell trapping, we obtained the optimal combinations of geometric parameters with a larger *Q* ratio difference before and after cell trapping: *W*_2_
*= H* = 25 μm, *W*_1_ = 7 μm, *L*_1_ = 12 μm, *R* = 50 μm.

To verify the performance of the optimized model obtained, three-dimensional structures with five cell capture sites were simulated to obtain the *Q* ratio when zero, one, two and three cells were trapped, respectively (Figure 4). Cells were treated as solid spheres with a diameter of 15 μm in contact with the constrictions (Figure 4D). From the flow velocity distributions of cross sections, it was observed that the flow velocity of the main channel increased remarkably after cell trapping. We also obtained the *Q* ratio values of the five sites of the optimized model when zero, one, two and three cells were trapped, respectively, as shown in Table 1. When a capture site was taken by one cell, the *Q* ratio of that site was found to drop below 1, losing the ability to capture a secondary cell, while the *Q* ratio of the next capture site remained higher than 1, retaining the ability to trap the next incoming single-cell. The *Q* ratios were consistent with the flow velocity distributions before and after cell trapping, which demonstrated the feasibility of the optimized structure for single-cell trapping.

Finally, fluid-structure interaction multiphysics coupling was used to simulate the cell movement trajectory of two-dimensional channels (Figure 5). Appendix A shows the simulation of cell movement trajectory. The white sphere represented a cell, and the different colors represented the different flow velocities. The simulation results showed that the flow velocity of the trapping channel was higher than the main channel before cell trapping, while the flow velocity of the main channel increased obviously after cell trapping. This agreed with the simulation results of three-dimensional channels and further suggested the feasibility of the designed chip for single-cell trapping theoretically.

### 3.4. Single-Cell Trapping and Immunofluorescence Staining Analysis

The single-cell trapping and immunofluorescence staining analysis were conducted to verify the feasibility of the single-cell trapping chip. Cell suspension and staining reagents were delivered into the chip by applying the pulling forces at the outlet. It was observed that cells were trapped deterministically, in which the cell flowed into the trapping channel and was captured at the capture site, and subsequent cell flowed around the trapping channel and flowed along the main channel to the next capture site. Appendix A shows the movement of cells in the chip, and the image of cells trapping is shown in Figure 6B. We obtained the single-cell capture efficiency for A549 and K562 of (82.7 ± 2.4)% and (84 ± 3.3)%,respectively (*N* = 3), as shown in Figure 6A. Furthermore, trypan blue staining demonstrated cells had good cell viability of 95% after cell trapping in the chip (Figure 6C).

Immunofluorescence staining is a common method for single-cell analysis due to its simple operation. However, immunofluorescence staining is a complex process that requires multistep washing and reagent addition. The reagent replacements required multiple intubations and re-intubations, which caused pressure fluctuations in the channels and resulted in cell movement and loss. Aiming to immobilize cells at the capture sites, we delivered the various staining reagents by applying a pulling force at the outlet with a syringe pump rather than a pushing force at the inlet. It was found that the cell remained in the capture positions and did not produce movement in the staining experiment. The cell fluorescence staining images of A549 cells are shown in Figure 7, in which the green fluorescence of CK-FITC and the blue fluorescence of DAPI are observed clearly under the inverted fluorescence microscope. The proposed immunofluorescence staining strategy resolved the cell movements and loss issues during the multistep immunofluorescence staining analysis without complex equipment, and provided a simple strategy for single-cell analysis.

## 4. Conclusions

A novel single-cell trapping microfluidic chip was developed based on the least flow resistance principle. To obtain a higher spatial and temporal efficiency, in this study the design allowed the main channel to produce higher flow resistance than the cell trapping channel by designing a large circular structure before the constriction to decrease the flow resistance of the trapping channel and designing a serpentine structure to increase the flow resistance of main channel, which directed cells into the capture sites and improved single-cell trapping efficiency. Numerical simulations were conducted to guide geometry choice and obtain the optimization model, confirming its feasibility for single-cell trapping theoretically. In addition, single-cell trapping followed by immunofluorescence staining was conducted and high-efficiency single-cell trapping was obtained. In addition, we reduced cell movement and loss in multistep immunofluorescence staining by applying the pulling force at the outlets to deliver the solutions sequentially, which helped to make cell immunofluorescence staining successful. The simple operation and high single-cell trapping efficiency gives the proposed microfluidic chip potential applications in single-cell trapping and biochemical analysis.

## Figures and Tables

**Figure 1 micromachines-13-01272-f001:**
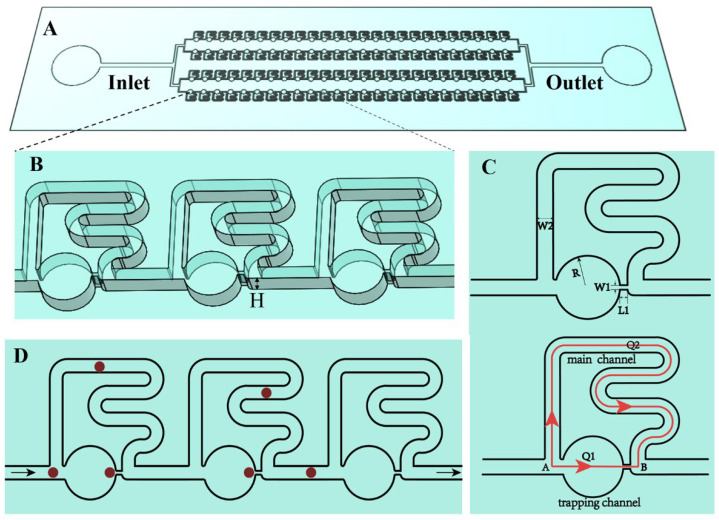
Microfluidic chip for single-cell trapping. (**A**) Schematic of the proposed single-cell trapping chip. (**B**) The detailed three-dimensional structure of proposed single-cell trapping chip. (**C**) The structure design of the single-cell trapping units. The geometric parameters *W1, L1, R, W2* affecting the trapping efficiency were labeled. The design of a large circular structure before the constriction and the serpentine structure allowed the main channel to produce higher flow resistance than the trapping channel and cells would flow along the trapping channel. (**D**) Schematic of cell movement in the channels. When the capture site was taken by the cell, the next cell would flow around the trapping channels and flow along the main channels to the next capture site.

**Figure 2 micromachines-13-01272-f002:**
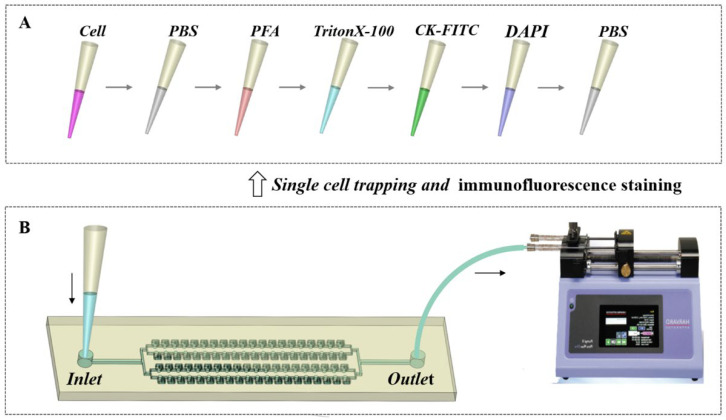
The procedure of single-cell trapping and immunofluorescence staining. (**A**) Cell suspension and immunofluorescence staining reagents were delivered into the chip sequentially. (**B**) The pulling force was applied at the outlet with a syringe pump to deliver the solution from the pipette tip into the chip.

**Figure 3 micromachines-13-01272-f003:**
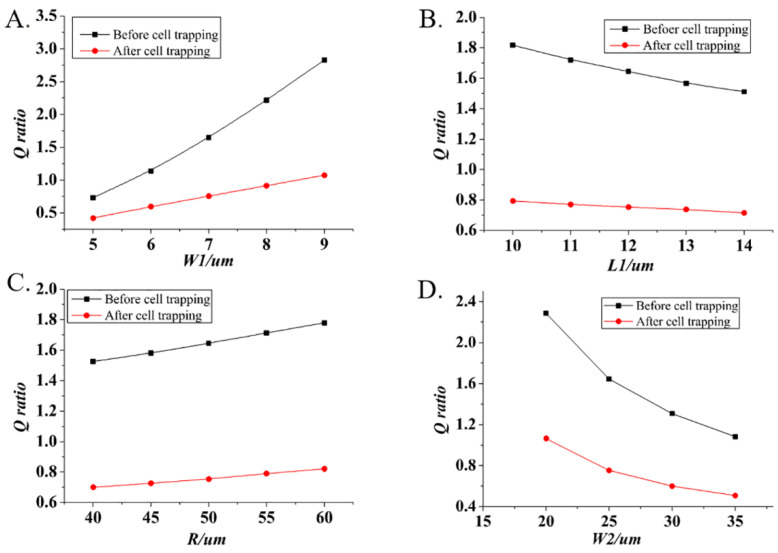
The effects of chip geometric parameters on the *Q* ratio values before and after cell trapping. (**A**) The *Q* ratio increased as *W_1_* increased before and after cell trapping. (**B**) The increase of *L*_1_ led to the decrease of *Q* ratio before and after cell trapping. (**C**) The *Q* ratio increased with the increase of as *R* before and after cell trapping. (**D**) The increase of *W_2_* caused the decrease of *Q* ratio before and after cell trapping.

**Figure 4 micromachines-13-01272-f004:**
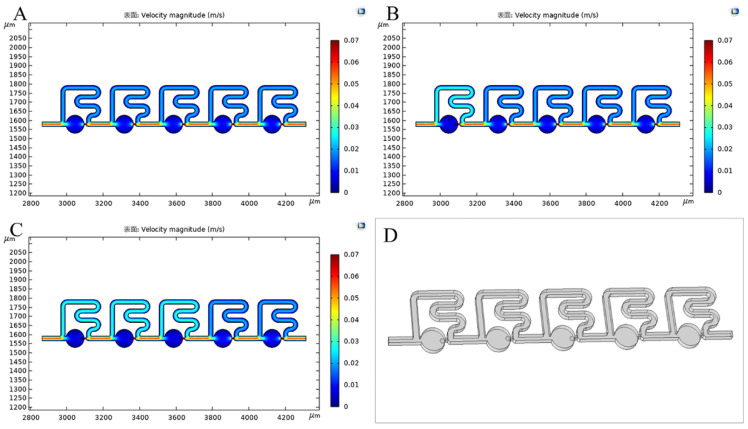
Simulations of flow velocity distributions of three−dimensional channels. (**A**) Flow velocity distributions before cell trapping. (**B**) Flow velocity distributions when one cell was trapped. (**C**) Flow velocity distributions when three cell capture sites were taken. (**D**) Three−dimensional channels used for the simulation, in which cells were assumed as solid spheres and took three capture sites.

**Figure 5 micromachines-13-01272-f005:**
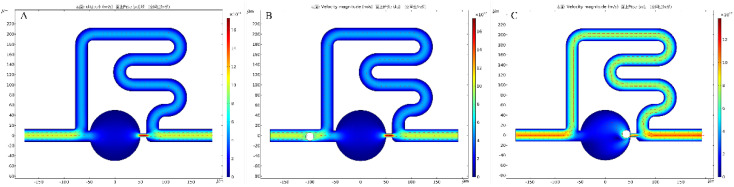
Simulations of flow velocity distributions of the two−dimensional channel. (**A**) Flow velocity distributions without cell in the channel. (**B**) Flow velocity distributions during the cell moved in the channel. (**C**) The flow velocity distributions after the cell was trapped at the capture site.

**Figure 6 micromachines-13-01272-f006:**
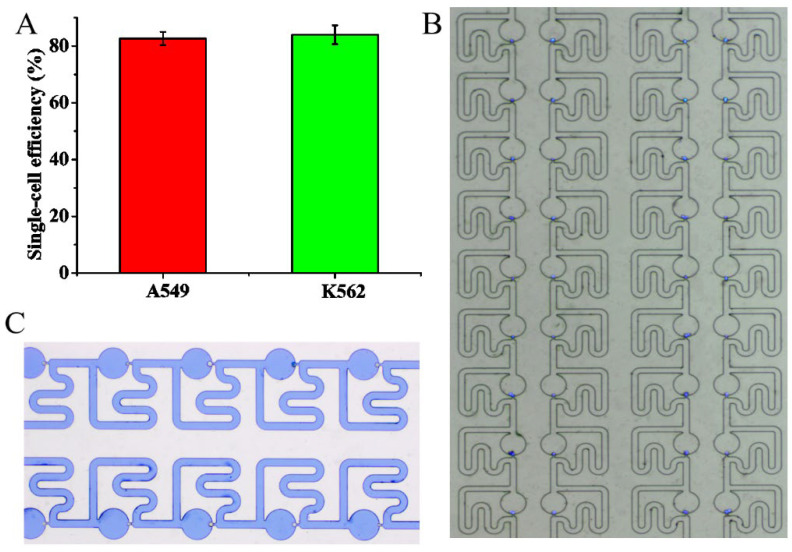
Single-cell trapping on the proposed chip. (**A**) Average single-cell trapping efficiency for A549 and K562 (*N* = 3 for each cell line). (**B**) The image of single-cell trapping and stained with DAPI, in which cells appeared as blue. (**C**) Measurement of cell viability using trypan blue staining. Trypan blue stained dead cells blue.

**Figure 7 micromachines-13-01272-f007:**
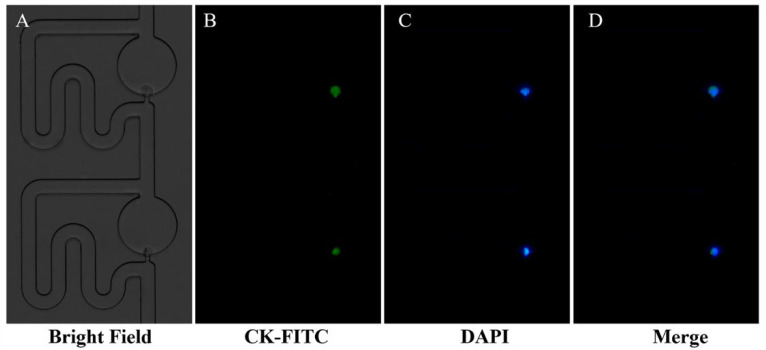
Immunofluorescence staining on the proposed chip. (**A**) Bright field image of the single-cell trapping. (**B**) Fluorescence images of cells stained with CK-FITC. (**C**) Fluorescence images of cells stained with DAPI. (**D**) Fluorescence merged images of cells stained with CK-FITC and DAPI.

**Table 1 micromachines-13-01272-t001:** Q ratio values of the five sites of the optimized model when zero, one, two and three cells were trapped, respectively.

Trap Site	1	2	3	4	5
No cell trapped	1.645	1.646	1.644	1.646	1.646
One	0.753	1.645	1.643	1.645	1.645
Two	0.753	0.757	1.643	1.645	1.645
Three	0.753	0.757	0.754	1.645	1.645

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
