# Peer review of "A Resistance-Based Microfluidic Chip for Deterministic Single Cell Trapping Followed by Immunofluorescence Staining"

_micromachines, 2022, doi:10.3390/mi13081272_

Round 1

Reviewer 1 Report

This paper presented a resistance-based cell capture microfluidic chip. The design and results were reasonable, because similar resistance-based cell trapping structures are very common, many notable results have been achieved. Therefore, the most important issue in this paper is its innovation. The authors declared that the existing methods were with “complicate and multilayer structure fabrication, expensive and bulky external instruments and low throughput”, which does not set up. The cell staining part is also a common design. So, the authors highly needed to clarify what innovation is. In addition, there are serious problems with the format of the paper, and even a lot of content that clearly does not belong to the paper.

Author Response

Thanks for the reviewer’s suggestion. We clarified the innovation of our chip and revised the format of the paper in revised the manuscript.

Reviewer 2 Report

This manuscript describes a microfluidic device for single-cell trapping. COMSOL simulation was employed for channel-design parameter optimization. A549 cells were trapped and stained on the chip as a proof of concept. The innovation of this work is in the design of the chip. The application of this technology may require more demonstration. The article writing is overall fine, however, the author may need to carefully check the experimental details that were provided in the manuscript. Also, more data is required to demonstrate the chip’s performance. 

Additional comments:

1. While the design of this chip is unique, some similar designs have already been reported. (e.g. Biomicrofluidics, 2018, 12(2): 024102.   Biomicrofluidics, 2015, 9(1): 014101.) Compared to those similar designs, what’s the advantage of this device that was described in this work?

2. The authors claimed the capture efficiency of this device is 88%. How was the capture efficiency calculated? 

3. According to the protocol, 6μL of 50 cells/μL solution was introduced into each channel, which was 300 cells overall. However, only 100 cell traps were equipped in each channel. Does it mean at least 2/3 loss occurs in each channel?

4. Is the capture efficiency relevant to the Q ratio? Will a higher Q ratio lead to a higher capture efficiency?

5. Will other parameters of the cell suspension solution, eg. Viscosity and contact angle, affect the capture efficiency?

6. The parameters of the geometry were optimized from the simulation result. However, the setting in detail was not provided.

7. Since the Q ratio can be affected by all geometry parameters, did the author consider finding the optimal condition via global optimization?

8. What was the concentration of CK-FITC used in the cell staining?

9. Can the trapped cell be released out of the chip? 

10. It would be nice if the author can provide more experimental data. For example, different concentrations of cells in the solution, different flow rates, etc.

11. What is the maximum size the chip can expand? (maximum number of cell traps)

12. It would be nice if the author can provide more experimental data with cells from different cell lines.

Author Response

Thanks for the reviewer’s comments. We revised the the manuscript according to the  reviewer’s comments using the “Track Changes” function of the word.

Reviewer 3 Report

In this manuscript, the authors reported a resistance-based method for single cell trapping by designing a large circular structure before constriction and the serpentine structure on the main channel. They also investigated the geometric parameters based on numerical simulations. Finally, they conducted immunofluorescence staining and obtained a single cell trapping efficiency of 88%. This work is useful for single cell tapping and analysis. However, I have several concerns before this manuscript can be accepted. Therefore, in its current form, revisions are needed.

1. Did the authors measure the viability of A549 cells? The authors should investigate the cell viability after trapping the cells.

2. Are there clogging and leaking issues during the experiment process and how did the authors avoid this?

3. Did the authors investigate the repeatability and stability of this chip?

4. Is it possible that the trapped cells can be released for further analysis using this microfluidic chip?

5. The authors should cite more references regarding the active and passive methods for single cell trapping. For example, 10.1021/jacs.7b03288, 10.1038/nmeth.1290, 10.1021/acs.analchem.8b02628.

Author Response

Thanks for the reviewer’s comments. We have revised the the manuscript according to the  reviewer’s suggestion.

Round 2

Reviewer 2 Report

I believe this manuscript is now ready to be published.

Reviewer 3 Report

The authors have addressed all my questions, and I think this manuscript is now acceptable for publication.